# Dynamic Translational Landscape Revealed by Genome-Wide Ribosome Profiling under Drought and Heat Stress in Potato

**DOI:** 10.3390/plants12122232

**Published:** 2023-06-06

**Authors:** Hongju Jian, Shiqi Wen, Rongrong Liu, Wenzhe Zhang, Ziyan Li, Weixi Chen, Yonghong Zhou, Vadim Khassanov, Ahmed M. A. Mahmoud, Jichun Wang, Dianqiu Lyu

**Affiliations:** 1College of Agronomy and Biotechnology, Southwest University, Chongqing 400715, China; hjjian518@swu.edu.cn (H.J.);; 2State Cultivation Base of Crop Stress Biology for Southern Mountainous Land of Southwest University, Chongqing 400715, China; 3Chongqing Key Laboratory of Biology and Genetic Breeding for Tuber and Root Crops, Chongqing 400715, China; 4Department of Plant Protection and Quarantine, Faculty of Agronomy, S. Seifullin Kazakh Agrotechnical University, Zhenis Avenue, 010011 Astana, Kazakhstan; 5Department of Vegetable Crops, Faculty of Agriculture, Cairo University, Giza 12613, Egypt

**Keywords:** drought stress, heat stress, potato, ribosome profiling, upstream open reading frames

## Abstract

The yield and quality of potatoes, an important staple crop, are seriously threatened by high temperature and drought stress. In order to deal with this adverse environment, plants have evolved a series of response mechanisms. However, the molecular mechanism of potato’s response to environmental changes at the translational level is still unclear. In this study, we performed transcriptome- and ribosome-profiling assays with potato seedlings growing under normal, drought, and high-temperature conditions to reveal the dynamic translational landscapes for the first time. The translational efficiency was significantly affected by drought and heat stress in potato. A relatively high correlation (0.88 and 0.82 for drought and heat stress, respectively) of the fold changes of gene expression was observed between the transcriptional level and translational level globally based on the ribosome-profiling and RNA-seq data. However, only 41.58% and 27.69% of the different expressed genes were shared by transcription and translation in drought and heat stress, respectively, suggesting that the transcription or translation process can be changed independently. In total, the translational efficiency of 151 (83 and 68 for drought and heat, respectively) genes was significantly changed. In addition, sequence features, including GC content, sequence length, and normalized minimal free energy, significantly affected the translational efficiencies of genes. In addition, 28,490 upstream open reading frames (uORFs) were detected on 6463 genes, with an average of 4.4 uORFs per gene and a median length of 100 bp. These uORFs significantly affected the translational efficiency of downstream major open reading frames (mORFs). These results provide new information and directions for analyzing the molecular regulatory network of potato seedlings in response to drought and heat stress.

## 1. Introduction

Abiotic stress can lead to crop-growth retardation or even death, and drought and heat stress are the main limiting factors that restrain crop growth and development [1]. Therefore, it is of great value to crop-resistance breeding to study the gene-expression regulation network in response to abiotic stress. Plant-protein stability [2,3] and cell-membrane fluidity [4,5] are greatly reduced when they are exposed to adverse environment stresses. To alleviate the negative effects of stress, plants can immediately regulate at the physiological, biochemical, and gene-expression levels. Plants respond to environmental stress through various mechanisms at the transcriptional and translational levels in maize [6]. Drought stress and heat stress have been studied comprehensively at the transcriptional level [7,8,9,10]. Transcriptional regulation is considered to be one of the most effective methods for plants to respond to environmental stress [11].

With the development of science and technology, it has been found that long-term exposure of plants to environmental stress can lead to genetic changes at the epigenetic level [12]. This regulation of the translational level is independent of the transcriptional level and plays an important role when plants face abiotic stress. Translation is a downstream step of transcription. When plants are subjected to environmental stress, the regulation of the translational level allows them to skip transcription and translate proteins with a lower concentration of RNA into higher concentration to cope with environmental stress [13]. This also explains the poor correlation between proteomic data and transcriptome data. At the same time, as the main carrier of translation, polyribosomes turn into mono-polyribosomes when plants are subjected to external stress, which suggests the existence of epigenetic regulation of translation inhibition in plants [14,15]. Among these translation-regulation mechanisms, some are highly similar between yeast and plants, suggesting that epigenetic translation regulation may be conserved during biological evolution [16]. However, due to the limitation of technology, the research on the regulation mechanism of the plant-translational level is lagging.

In recent years, a technique called ribosome profiling has been developed to obtain complete data genome wide, with single-codon resolution in translation processes by deep sequencing of ribosome-protected fragments (RPFS) [17,18]. Ribosome profiling quantizes the process at the level of genome-wide translation. As an important breakthrough in the development of modern molecular biology, ribosome profiling has been widely used in the study of mammalian translation-regulatory networks, such as thermal-stress response [19], proteotoxic stress [20], oxidative stress [21], etc. Ribosome profiling has also been used to study the translation-regulation mechanism of *E. coli* [22] and yeast [23]. As a bridge to connecting transcriptomics and proteomics, ribosome profiling was employed in plants, revealing the regulation mechanism of different translational levels under heat stress [19], light [20], and oxidative stress [21]. Most translational-omics studies have shown that plants and animals promote or inhibit translation via the readjustment of ribosome-distribution density on mRNA [19,21]. It can affect also the translational efficiency of genes by mediating ribosome re-selection of translation initiation, scanning speed, code-shifting mutation, and codon read-through [24]. In addition, many studies have shown that the upstream open reading frame plays a role in promoting or inhibiting gene translation in gene-translation regulation [6,24,25].

In our study, we investigated and elucidated the mechanism of the regulation response of potato under high temperature and drought by combining translational omics and transcriptomics. The information obtained in this study will provide a new direction to improve tolerance to high temperature and drought stress in potato.

## 2. Results

### 2.1. Characteristics of Ribosome-Profiling Data

In order to systematically study the response of potato to drought and heat stress at the transcriptional and translational levels, RNA-seq and ribosome profiling were performed on drought-stress (noted as “D”), heat-stress (noted as “H”), and control (noted as “CK”) potato seedlings with two biological replicates in this study. As shown in Appendix A, the plant status was affected after being subjected to drought and high-temperature stress. *StRD29B1* and *StHSP101*, marker genes for drought and heat stress, respectively, were significantly induced (Appendix A), which indicates that the plant was damaged after stress. In total, around 78, 76, and 86 million ribosome-profiling reads were generated in the CK, D, and H potato seedlings, respectively (Appendix A). Furthermore, 93,124 and 109 million RNA-seq reads were obtained in the CK, D, and H potato plants, respectively (Appendix A). As described in the Section 4, the adapter and low-quality sequences in the raw reads of the ribosome profiling were removed and then the rest clean reads were aligned to the GenBank and Rfam database (Appendix A). After removing the sequences mapped to rRNAs, tRNAs, snoRNAs, snRNAs, miRNAs, and other snRNAs, the rest of the 25–35 nt reads were then mapped to the potato reference genome (DM 1-3 516 R44). The high correlation coefficient between the two biological replicates of the transcriptome and translatome indicates the repeatability and reliability of our data (Appendix A).

To further analyze the impact of drought and heat stress on the characteristics of ribosome-profiling data, the basic ribosome profiles of RPFs among CK, D, and H samples were then compared. The majority of the length distribution of RPFs was around 33 nt in all six samples (Figure 1a), slightly longer than the 28 nt in yeast [26], 30 nt in Arabidopsis [20] and maize [6], and 32 nt in tea plant [25]. Recently, small ORFs in the untranslated regions (UTRs) and long noncoding RNAs (lncRNAs) were identified among several eukaryotes, such as yeast [27], mammals, and plants [28,29]. The laws of RPF distribution ratios were similar among CK, D, and H samples. Briefly, most RPFs were mapped on the CDS region (~80%), followed by 3′UTR (~11%), whereas 5′UTR and intron had much lower distributions of RPFs (Figure 1b). The distribution patterns of RPFs in this study were consistent with those in other species, confirming the high quality of the Ribo-seq library (Lei, 2015). The proportion of RPFs mapped to CDS, intron, and UTR regions were similar among CK, D, and H samples (Figure 1b). A strong three-nucleotide periodicity was observed around the start codon and stop codon (Figure 1c).

### 2.2. Expression Changes at the Transcriptional and Translational Levels in Response to Drought and Heat Stress

Drought affects plant growth and development at the molecular and physiological levels. We were able to screen the DEGs at the transcriptional and translational levels simultaneously using RNA-seq and ribosome-profiling data with the following criterion: fold changes > 2 and FDR < 0.05. For drought stress, there were 2345 up-regulated genes and 2048 down-regulated genes at the transcriptional level and 1227 up-regulated genes and 1248 down-regulated genes at the translational level (Figure 2a). Among these DEGs, 1033 up-regulated and 984 down-regulated DEGs were coordinately regulated both at the transcriptional and the translational level (Figure 2b). For heat stress, 2295 DEGs (992 up-regulated genes and 1267 down-regulated genes) and 798 (394 up-regulated genes and 404 down-regulated genes) were detected at the transcriptional and translational levels, respectively (Figure 2c). Meanwhile, 341 up-regulated and 322 down-regulated DEGs were regulated at both the transcriptional and the translational level (Figure 2d). Significantly, the IGV browser view displaying global RNA-seq and Ribo-seq of 12 chromosomes indicates that drought and heat stress affected significant changes at the transcriptional level, the translational level only (highlighted in light-green shading), and at both levels (shaded in light red, Figure 2e). To test the accuracy of the transcriptome results, 13 DEGs were randomly selected using qRT-PCR to confirm the reliability of the RNA-seq data. Similar change trends between the transcriptome and qRT-PCR were detected, which indicates the accuracy of our sequencing data (Appendix A).

In order to further analyze the expression trend of genes at different levels, we calculated the fold changes of all genes at both the transcriptional and translational level. There was a relatively high correlation between the transcriptional and translational levels (R^2^ = 0.88 and 0.82 in drought and heat stress, respectively, Figure 3). Based on the criteria of |log 2 [fold change]| ≥ 1 and FDR < 0.05 parameters, the change patterns of these genes can be divided into nine categories (Figure 3a,c). For drought stress, more than 77.17% (10,277) of genes showed no significant change in gene expressions at either the transcriptional or the translational level (quadrant E), whereas only 12.49% (1667) of genes were regulated congruously at both the transcriptional and translational levels (up-regulation for quadrant C; down-regulation for quadrant G) and the remaining 10.34% (1374) of genes in the other six groups (quadrants A, B, D, F, H, and I) were discordantly regulated at the transcriptional and translational levels (Figure 3a). Genes in all nine groups except the A and I quadrants were analyzed by KEGG annotation and enrichment (Figure 3b). Genes in quadrant D (transcriptionally down-regulated with no significant change in translation) were enriched in nine pathways, such as ribosome, carbon metabolism, fructose, and mannose metabolism, whereas five pathways, such as starch and sucrose metabolism and ether-lipid metabolism were enriched in quadrant F (transcriptionally up-regulated with no significant change in translation). In contrast, genes in quadrants B and H were only regulated at the translational level, with no significant changes at the transcriptional level and not significantly enriched in any pathways. For heat stress, more than 81.51% (10,480) of genes were in quadrant E were not significantly regulated at either the transcriptional or the translational level, whereas only 7.48% (961) of genes were regulated congruously at both levels and 14.35% (1416) of genes were located in the remaining six quadrants (Figure 3c). KEGG analysis revealed that nine pathways, including biosynthesis of secondary metabolites, biosynthesis of amino acids, and carbon metabolism, were enriched in quadrant D, whereas protein processing in the endoplasmic reticulum was significantly enriched (q-value = 6.2 × 10^−38^) in quadrant C and six pathways, including phenylalanine metabolism, phenylpropanoid biosynthesis, biosynthesis of secondary metabolites, and plant-hormone signal transduction, were enriched in quadrant G (Figure 3d). Interestingly, encoding ABA core-signal-component genes *PP2CA* (*PGSC0003DMG400016742*), *ABF2* (*PGSC0003DMG400008011*), and *ABF4* (*PGSC0003DMG400022931*) were up-regulated at both the transcriptional and translational level after drought stress, and heat-shock-factor gene *HSF30* (*PGSC0003DMG400008223*) and heat-shock-protein gene *HSP18.2* (*PGSC0003DMG400011632*) were significantly up-regulated after heat stress at both levels (Figure 4). Aquaporins gene *PIP1-1* (*PGSC0003DMG400020742*) was down-regulated at both levels after drought and heat stress (Figure 4).

### 2.3. The Translational Efficiencies of a Large Number of Genes Were Significantly Changed in Response to Drought and Heat Stress

As the ratio of the FPKM value of ribosome profiling to RNA-seq, translational efficiency (TE) is an important indicator of translation, reflecting the efficiency of RNA utilization [26,30]. The genome-wide TE analysis showed that log_2_ (TE) values were mainly concentrated between –5 and 5, indicating that different genes had different translation-regulation modes (Figure 5). In addition, the TE of a small number of genes increased by more than 1000 times, whereas other genes were reduced ~60-fold, which shows that these genes underwent different modes of regulation (Figure 5). There were 83 and 68 genes with significantly changed TE in drought and heat stress, respectively, in comparison with the control sample, for which the number of genes with significantly changed TE was much lower than at either the transcriptional or the translational level (Appendix A).

### 2.4. Gene-Sequence Features Significantly Affect the Translational Efficiencies

Gene-sequence characteristics are considered to significantly affect protein abundance [31,32,33]. In order to further analyze the contribution of potato gene-sequence features to protein translational efficiency, we detected the effects of sequence length, normalized minimal free energy (NMFE), and GC content in 5′UTR, CDS, and 3′UTR of the corresponding genes on translational efficiency. In the CK sample, there was little difference between higher and lower TE-group genes, and only the lowest TE group (Log 2 (TE) ≤ −1) had a shorter length, lower GC content, and higher NMFE in 3′UTR (Figure 6a). For 5′UTR, shorter length, higher GC content, and higher NMFE were detected (Figure 6b), whereas longer length, higher GC content, and lower NMFE in the CDS region were observed in lower TE (Log 2 (TE) ≤ −1) genes (Figure 6c). The gene-sequence characteristics in different TE groups had a similar pattern in control, drought- (Appendix A), and heat-stress (Appendix A) samples. Then, we detected whether genes with accordantly and discordantly expressed patterns varied in their sequence features. The cumulative curve showed that genes located in the accordant group had higher NMFE in the CDS regions compared with those in the discordant one under both drought (*p*-value = 0.029) and heat stress (*p*-value = 0.069), whereas the remaining regions were statistically indistinguishable (Appendix A).

### 2.5. Characterization of uORFs and Their Impact on the Translation of mORFs

UTRs, especially 5′UTR, are involved in the post-transcriptional-regulation process [34]. Drought and heat stress altered the RPF proportion in 5′UTR (Figure 1), and these results indicated that there may be some regulatory elements involved in drought- or heat-stress response in 5′UTR. It has previously been reported that uORFs, small open reading frames in 5′UTRs, can significantly influence the translation of downstream main ORFs (mORFs) [35]. However, little is currently known about uORFs of genes in potato and their functions in response to drought or heat stress. In this study, we performed a genome-wide scale in potato to identify uORF information using the ribosome-profiling data. The results indicate that, based on the presence of AUG start codons in 5′UTR sequences, 6463 genes were expected to have 28,490 uORFs, with an average of 4.4 uORFs per gene and a median length of 100 bp (Appendix A, Figure 7a,b). In total, 9180, 9208, and 9087 translated uORFs (FPKM ≥ 1) were detected in the CK, D, and H samples, respectively (Appendix A). These results were consistent with the length distribution of uORF in other higher plants, such as Arabidopsis, and rice [36].

To depict the difference in sequence characteristics between translated and untranslated uORFs, we analyzed three parameters related to the re-initiation of the mORF, namely, uORF length, 5′ UTR length, and NMFE [29,37]. The lengths of the translated uORFs (*p*-value < 1.06 ×10^−60^, Figure 7c) and the 5′ UTR (*p*-value < 4.1 × 10^−3^, Figure 7d) of genes were significantly longer than those in untranslated ones in the CK samples, and they all displayed similar folding potential (Figure 7e). In addition, we observed that the relative distances from the uORF to the mORF start codon (*p*-value = 0.29, Figure 7f) and from the uORF to the transcription start site (TSS) (transcription start site; *p*-value < 2.4 × 10^−2^) in translated uORFs were shorter than in untranslated ones (Figure 7g). The differences in sequence features between translated and untranslated uORFs in D (Appendix A) and H (Appendix A) samples were consistent with those in the CK samples. As sequence features around the AUG start codon significantly affect the start-codon recognition process and translation initiation, we further compared the frequency of bases around the uORF start codon between translated and untranslated regions. This is of critical important for start-codon recognition and translation initiation in the position of −3(A/G) and 4(G) around the AUG start codon [38,39]. Then, we checked the Kozak consensus sequences of the translated uORFs and their main ORFs. As expected, this conserved pattern among the downstream main ORFs was observed, whereas this pattern was missing in the uORF (Figure 7h, Appendix A). Similar results were also reported in Arabidopsis [20] and tomato [40]. To further analyze the influence of the translated uORFs on the TE of the mORFs in potato, we determined the changes in the TE in three sets of genes, including genes with untranslated uORFs, genes with one, and genes with multiple translated uORFs. However, no significant changes were detected among the three sets of genes (Figure 7i, Appendix A). uORFs were reported as key regulators involved in plant development and environmental-stress response. Compared with the CK sample, TE of uORFs was significantly changed in the D (*p* < 5.02 × 10^−4^) and H (*p* < 3.26 × 10^−3^) samples (Figure 7j), which is consistent with the changes in the proportion of RPFs in UTRs (Figure 1b).

## 3. Discussion

To date, many omics techniques have been used to reveal the response strategies of plants to abiotic stress. However, the response of crops, especially potato, to drought and high-temperature stress at the translational level remains to be analyzed. In this study, a combination of RNA-seq and ribosome-profiling technologies was used to explore the drought- and high-temperature-regulation landscapes of potato leaves at the transcriptional and translational levels. Our results aim to provide new ideas regarding the response and regulation of potato under drought and high-temperature stress in terms of multiple aspects.

### 3.1. Transcriptional and Translational Regulation Coordinate to Respond to Drought and Heat Stress

As a highly dynamic and precisely regulated process, translation is an important way for plants to cope with adverse environments [41]. However, in many cases, mRNA and corresponding protein levels are not correlated [25]. Therefore, scientists should pay attention to this process. In this study, the transcriptional and translational levels were significantly changed in potato after drought and high-temperature stresses (Figure 2). Relatively high Pearson correlation coefficients (0.88 and 0.82 for drought and high-temperature stress, respectively) between the fold changes of gene expression of transcription and translation were observed (Figure 3). Translation regulation does not need to generate new messenger RNA, so it is a fast and direct way to cope with adverse environments [42]. There were 1374 (45.2%) and 1416 (59.6%) genes under drought and heat stress, respectively, that underwent inconsistent change at the transcriptional or translational levels, which clearly suggests various stress responses at the two levels (Figure 3). Of them, 597 and 661 discordant regulated genes under drought and heat stress, respectively, were exclusively regulated at the translational level (Figure 3), whereas 775 and 754 genes (drought and heat stress, respectively) in groups D and F were exclusively changed at the transcriptional level (Figure 3). As reported in yeast, the mRNA abundance at the time point is more closely related to the protein abundance at the next time point than at the same time point, indicating that the protein expression lags behind transcription in response to adverse environments [43]. Therefore, we assume that the changes in the mRNA abundance of the genes in groups D and F can be used as indicators of subsequent translation changes.

### 3.2. TE Was Significantly Affected by Sequence Features

In general, gene expression is precisely regulated by TFs, microRNAs, lncRNAs, etc. In addition, it is also affected by its own sequence characteristics. In this study, TE varied with different sequence features in both 3′UTR, 5′UTR, and CDS regions in the three samples (Figure 6, Appendix A). Several studies revealed that there are many translational-regulatory elements in the UTRs of genes. uORFs, for example, play a critical role in translation repression [20]. However, inconsistent with previous studies [6,18,44], no significant changes were observed to indicate that potato uORFs affected the TE of mORFs (Figure 7i, Appendix A). In addition, the TE of the translated uORFs was significantly decreased with the increasing number of RFs in the 5′ UTR in the D sample, whereas the TE of the translated uORFs was significantly increased with the decreasing number of RFs in the 5′ UTR in the H sample (Figure 1b and Figure 7j). These findings suggested that potato decreased gene-expression levels by increasing the translation of uORFs under drought or heat stress. Furthermore, we also compared sequence features of translated and untranslated uORFs (Figure 7c–h, Appendix A), and these differences were closely related to the recognition of the initial codon of uORF and the translation initiation of mORF [29]. Furthermore, mORFs were more conserved with the Kozak sequence in the sequence flanking ATG than that in uORFs of both translated and untranslated uORFs, which were least conserved (Figure 7h, Appendix A). A previous study revealed that modifying the uORF of *LsGGP2*, a key enzyme in vitamin C biosynthesis in lettuce, increased both oxidation-stress tolerance and ascorbate content [45]. Therefore, the uORFs detected in this study provide new directions for improving drought or heat tolerance in potato.

## 4. Materials and Methods

### 4.1. Plant Materials

Tissue-culture seedlings of potato variety “Desiree” were used in this study. The seedlings were grown on MS medium in a light incubator at Chongqing Key Laboratory of Biology and Genetic Breeding for Tuber and Root Crops. The culture conditions were 22 °C, light for 16 h (3000 lx), and dark for 8 h. After growth for 14 days, these seedlings were transferred to hydroponic solution containing quarter-strength modified Hoagland nutrient solution. After 7 days, these seedlings were randomly divided into three groups for treatment of drought stress, heat stress, and control. For drought stress, 20% PEG-6000 was added to the solution and 20 leaves were harvested from at least 10 plants after 3 h treatment. For heat stress, 20 tissue-culture seedlings were moved to another incubator at 35 °C and with light for 16 h (3000 lx) and dark for 8 h conditions, and 20 leaves from at least 10 plants after 3 h were harvested. For CK samples, 20 leaves from at least 10 plants were harvested. The experiments were repeated twice, and all samples were immediately frozen in liquid nitrogen and stored at −80 °C until use.

### 4.2. Ribosome Profiling

One milligram of leaf powder was dissolved in 400 µL lysis buffer (containing Tris-HCL (pH 7.8, 5 mM), MgCL_2_ (10 mM), cycloheximide (100 μg/mL), and H_2_O) with cycloheximide, and the re-suspended extracts in lysis buffer were transferred to new microtubes, pipetted several times, and incubated on ice for 10 min. Then, the cells were triturated 10 times using a 26 G needle and the supernatant was collected after centrifuging at 20,000× *g* for 10 min at 4 °C. To prepare ribosome footprints (RFs), 7.5 µL RNase I (100 U/μL) and 5 µL DNase I were added to 300 µL of lysate to incubate for 45 min at room temperature with gentle mixing on a Nutator mixer. Nuclease digestion was stopped by adding 10 µL RNase inhibitor. To recover ribosomes, size-exclusion columns (illustra MicroSpin S-400 HR Columns; GE Healthcare; catalog no. 27-5140-01) were equilibrated with 3 mL of polysome buffer and centrifuged at 600× *g* for 4 min at room temperature. Then, 100 μL digested RFs were added to the column and centrifuged at 600× *g* for 2 min and 10 μL 10% SDS was added to the elution, and RFs with a size greater than 17 nt were isolated according to the RNA Clean and Concentrator-25 kit. For rRNA depletion, a Ribominus rRNA Depletion Kit (Thermo Fisher, Waltham, MA, USA) was used to delete the rRNA. After obtaining ribosome footprints as indicated above, NEBNext^®^ Multiple Small RNA Library Prep Set for Illumina^®^ (catalog no. E7300S, E7300L) was used to construct Ribo-seq libraries. Briefly, adapters were added to both ends of RFs, followed by reverse transcription and PCR amplification. The 140-160 bp PCR products were enriched to generate a cDNA library and sequenced using Illumina HiSeq TM 2500 by Gene Denovo Biotechnology Co. (Guangzhou, China). All raw data were deposited in the National Center for Biotechnology Information (NCBI) sequence-read archive with accession number PRJNA922705.

### 4.3. mRNA-Sequencing-Library Construction and Sequencing

TRIzol reagent (Invitrogen) was used to extract the total RNA of each sample from CK, D, and H, and the manufacturer’s protocol of the Illumina Standard mRNA-seq library-preparation kit (Illumina, http://www.illumina.com/, accessed on 26 March 2023) was used to prepare the RNA-seq libraries. Then, RNA-seq libraries were sequenced on an Illumina HiSeq 2500 platform with paired-end technology (2 × 150) by Gene Denovo Biotechnology Co. (Guangzhou, China). To obtain high-quality clean reads, low-quality reads, adaptor sequences, and duplication sequences were removed. Low-quality reads were filtered by fastp. Raw reads containing more than 50% of low-quality bases or more than 10% of N bases were removed. Adapter sequences were trimmed. Reads with lengths between 20 and 40 bp were retained for subsequent analysis. The short-read alignment tool Bowtie2 was used for mapping reads to the ribosome RNA (rRNA) database, GenBank, and Rfam database. Reads mapped to rRNAs, transfer RNAs (tRNA), small nuclear RNAs (snRNA), small nucleolar RNAs (snoRNA), and miRNA were also removed. Then, the clean reads were mapped to the reference genome of DM 1-3 516 R44 using HISAT2.2.4. The FPKM (Fragments Per Kilobase of transcript per Million mapped reads) method was used to quantify gene abundance, and DEG identification was performed using DESeq2 [46] software with the following parameter: FDR < 0.05 and absolute fold change ≥ 2. GO and KEGG were also conducted to annotate gene functions and enriched metabolic pathways. All raw data were deposited in the National Center for Biotechnology Information (NCBI) sequence-read archive with accession number PRJNA925814.

### 4.4. Ribo-seq Data Analysis

Before further data analysis, low-quality reads and adapter sequences were filtered using fastp (version 0.1) [47]. The remaining reads with lengths between 10 and 50 bp were then mapped to rRNAs, tRNA, small nuclear RNA (snRNA), and small nucleolar RNA (snoRNA) databases. In addition, the unmapped reads with lengths between 25 and 35 bp were retained and mapped to the reference genome of DM1-3 using Bowtie2 [48] with no mismatches. RFs were assigned to different genomic features (5′UTR, CDS, 3′UTR, and others) based on the position of the 5′ end of the alignment. Furthermore, the RF density at different codon positions was calculated to monitor sequencing reliability.

### 4.5. Calculation of Translational Abundance and Calculation of Translational Efficiency (TE)

To calculate translational abundance, the read number in the open reading frame of coding genes was calculated by software RiboTaper [49], and the gene-expression level was normalized using the FPKM method. TE was calculated by FPKM at the translational level and at the transcriptional level of the above genes with FPKM ≥ 1. Moreover, the TE distribution of genes was shown as Log2-transformed TEs.

### 4.6. Correlation between Transcriptional and Translational Abundance

To make a comparison between transcriptional and translational abundance, genes with FPKM ≥ 1 were used to calculate the pairwise Pearson correlation (R^2^). According to the expression changes at the transcriptional level and translational level, the genes were classified into nine groups: A (transcriptionally down-regulated and translationally up-regulated genes), B (transcriptionally unchanged and translationally up-regulated genes), C (transcriptionally and translationally up-regulated genes), D (transcriptionally down-regulated and translationally unchanged genes), E (transcriptionally and translationally unchanged genes), F (transcriptionally up-regulated and translationally unchanged genes), G (transcriptionally and translationally down-regulated genes), H (transcriptionally unchanged and translationally down-regulated genes), and I (transcriptionally up-regulated and translationally down-regulated genes). Enrichment analysis of the GO functions and KEGG pathways was performed for all genes in these nine groups. Group-C and -G genes were defined as accordantly regulated genes and the genes in the remaining groups were defined as discordantly regulated. Integrative Genomics Viewer (IGV) Version 2.8.3 was used to visualize the expression changes at the transcriptional and translational levels simultaneously.

### 4.7. The Analysis of the Sequence Features among Four Groups of Different Translational Efficiency

Genes with different translational efficiency were artificially divided into four groups (TE ≤ −1, −1 < TE ≤ 0, 0 < TE ≤ 1, and TE ≥ 1). Normalized minimal free energy (NMFE) was one of the characteristic indices of the sequence stability of the secondary structure, which was predicted using the software RNAfold and normalized by the sequence length [50]. The *p*-value of the difference in gene features between the two TE groups were calculated using the two-tailed Student’s *t*-test (a = 0.05).

### 4.8. Analysis of uORFs

5′ UTR of known protein-coding genes with lengths ranging from 60 bp to 460 bp was extracted to predict uORFs. FPKM ≥ 1 was used as a parameter to determine whether it was a translated uORF, and R library (SeqLogo) was used to display the motif enrichment around the start codon between translated and untranslated uORFs (Bembom, 2007).

### 4.9. qRT-PCR Validation

One mg total RNA from CK, D, and H samples was used to synthesis cDNA using the Transcriptor First-Strand cDNA Synthesis Kit (Roche, Basel, Switzerland) and qRT-PCR was conducted on a Bio-Rad CFX96 Real-time System with SYBR Green PCR Supermix (CA, USA) as described previously [51]. *StRD29B1* and *StHSP101* were selected as marker genes for drought and heat, respectively. Another 13 DEGs were randomly selected to confirm the reliable of RNA-seq data. *StEF1α* was used as a control. All primers used in this study are shown in Appendix A.

## Figures and Tables

**Figure 1 plants-12-02232-f001:**
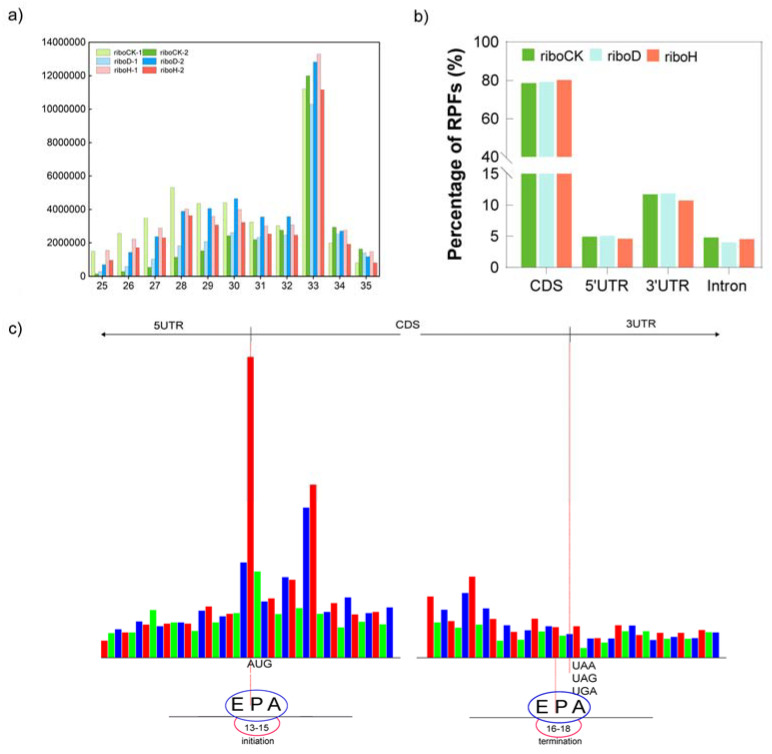
Characteristics of ribosome-profiling data in CK, D, and H samples. (**a**) Length distribution of RPFs in CK, D, and H samples; (**b**) the percentage of RPFs mapped on CDS, 5′ UTR, and 3′ UTR in CK, D, and H samples; (**c**) meta-analysis of all nucleotide-ribosome footprints near the start and end sites of annotation translation defined in ITAG3.2. The red, green, and blue bars represent reads mapped to the first, second, and third read frames, respectively.

**Figure 2 plants-12-02232-f002:**
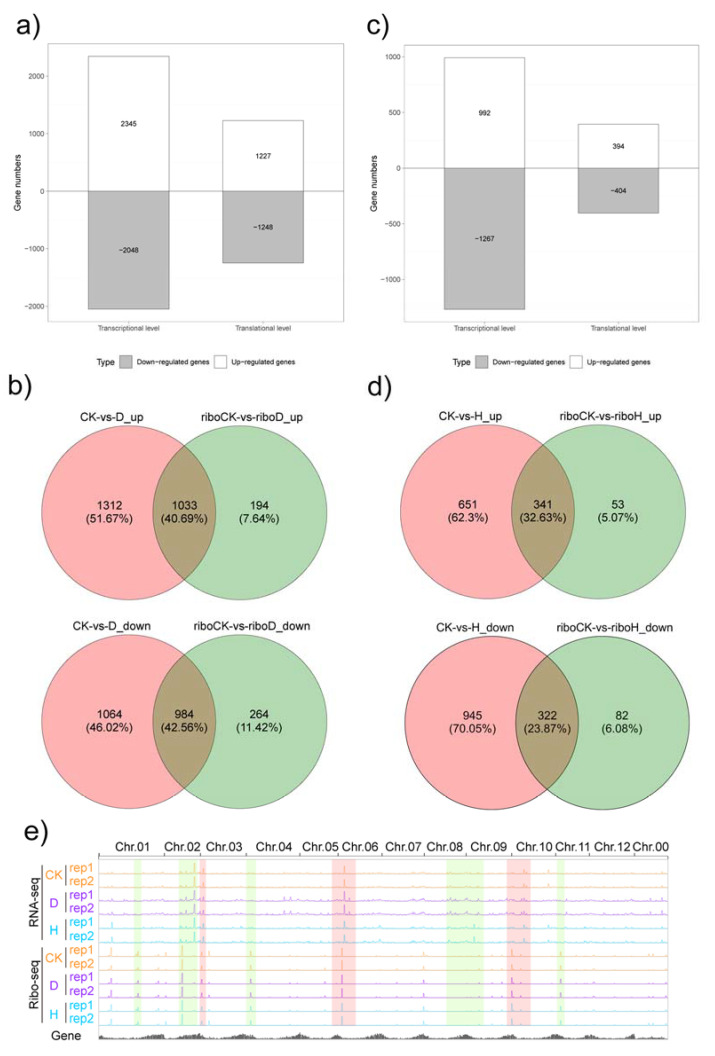
Transcriptional and translational regulation after drought or heat stress in potato. (**a**) The number of DEGs (fold change > 2 and FDR < 0.05) at the transcriptional and translational levels after drought stress; (**b**) Venn analysis of DEGs at the transcriptional and translational levels after drought stress; (**c**) the number of DEGs (fold change > 2 and FDR < 0.05) at the transcriptional and translational levels after heat stress; (**d**) Venn analysis of DEGs at the transcriptional and translational levels after heat stress; (**e**) IGV browser view displaying global RNA-seq and Ribo-seq tracks after drought or heat stress. Green highlighted boxes indicate that enrichment occurs in RNA-seq or Ribo-seq. Red highlighted boxes indicate co-enrichment in RNA-seq and Ribo-seq.

**Figure 3 plants-12-02232-f003:**
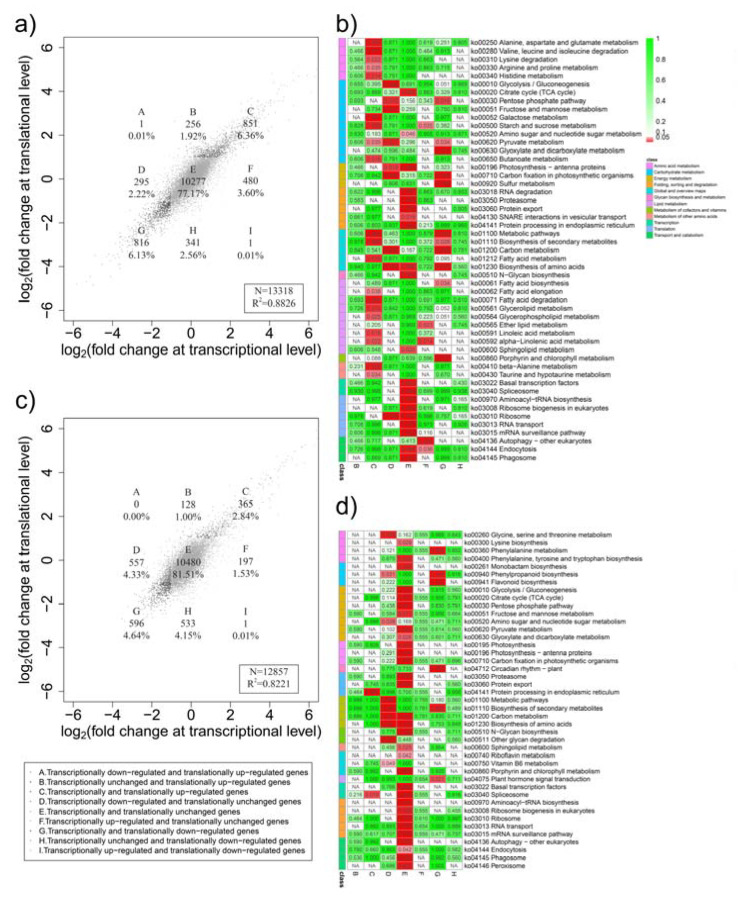
Transcriptional and translational changes in potato after drought or heat stress. (**a**) Scatter plot of fold changes at the transcriptional and translational levels after drought stress; (**b**) KEGG enrichment analysis of drought-responsive genes in nine groups except A and I; (**c**) scatter plot of fold changes at the transcriptional and translational levels after heat stress; (**d**) KEGG enrichment analysis of heat-responsive genes in nine groups except A and I.

**Figure 4 plants-12-02232-f004:**
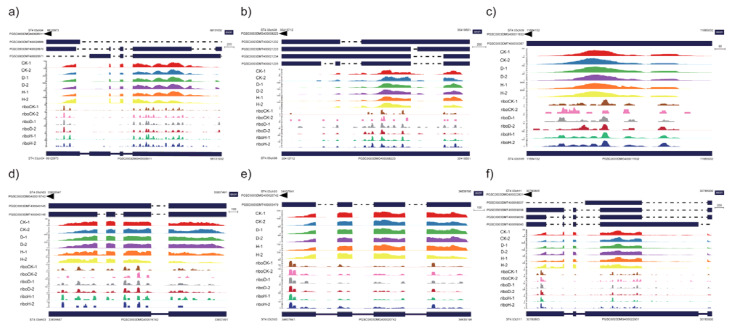
IGV browser view showing the changes of genes at the transcriptional and/or translational level after drought and heat stress in potato. (**a**) *PP2CA* (*PGSC0003DMG400016742*); (**b**) *ABF2* (*PGSC0003DMG400008011*); (**c**) *ABF4* (*PGSC0003DMG400022931*); (**d**) *HSF30* (*PGSC0003DMG400008223*); (**e**) *HSP18.2* (*PGSC0003DMG400011632*); and (**f**) *PIP1-1* (*PGSC0003DMG400020742*).

**Figure 5 plants-12-02232-f005:**
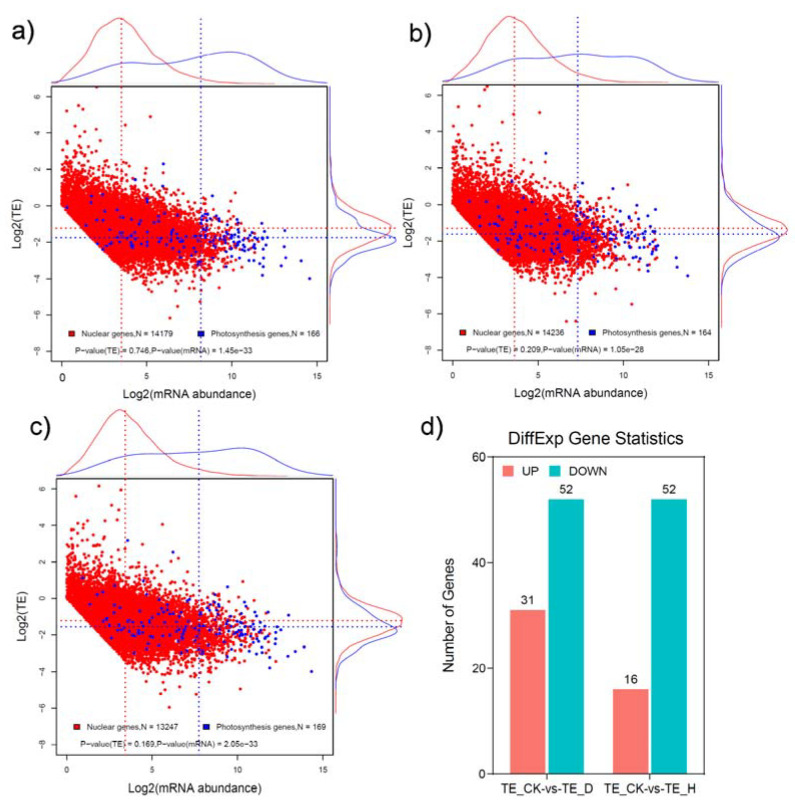
Genome-wide translational-efficiency (TE) analysis in CK, D, and H samples. (**a**) An overview of the log 2 (TE) in the CK sample; (**b**) an overview of the log 2 (TE) in the D sample; (**c**) an overview of the log 2 (TE) in the H sample. Blue and red colors show photosynthesis genes and nuclear genes, respectively. The blue and red dashed lines show the mean value of the corresponding genes. *p*-values were tested by the single-tailed Student’s *t*-test. (**d**) Number of differentially expressed TEs after drought or heat stress.

**Figure 6 plants-12-02232-f006:**
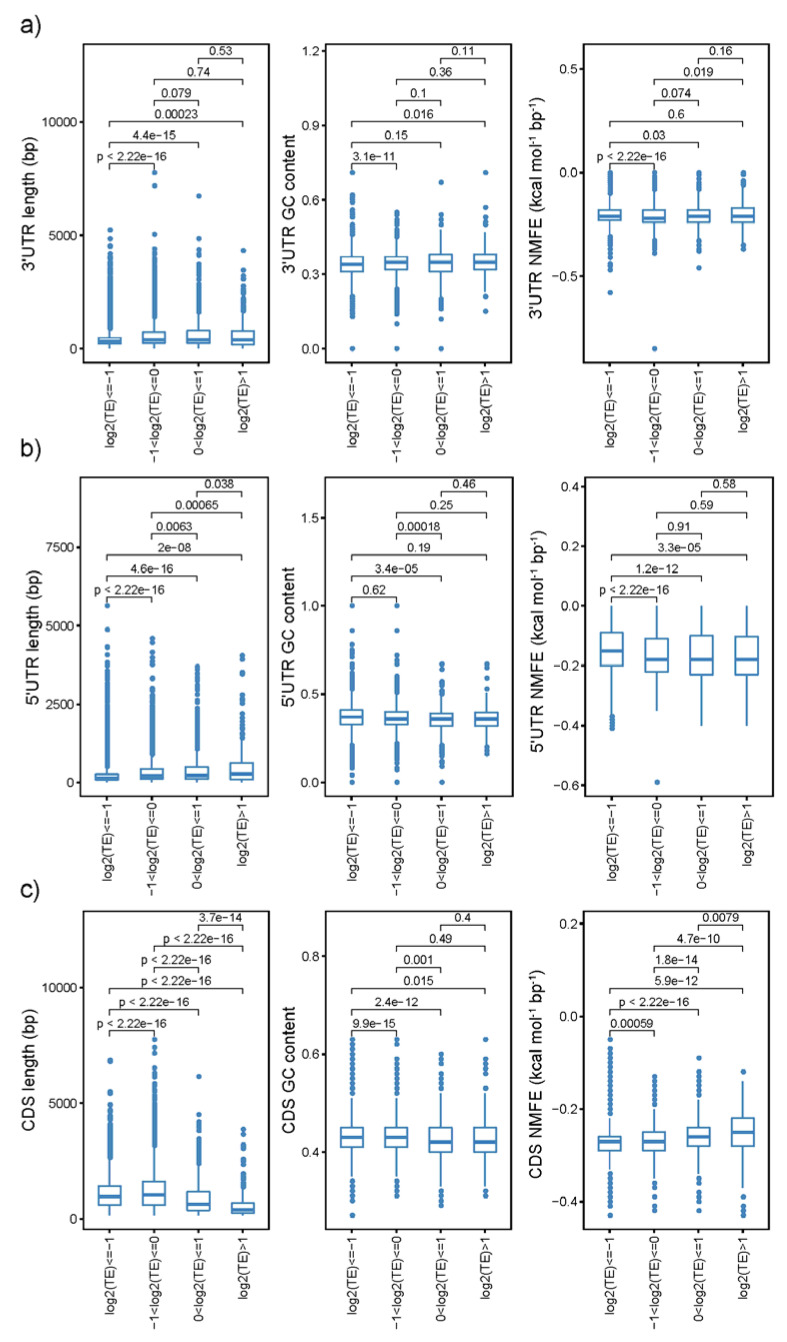
The variance of sequence length, GC content, and NMFE among four TE groups in the 3′UTR (**a**), 5′UTR (**b**), and CDS (**c**) regions of CK samples. The letters indicate significant differences based on Student’s *t*-test (*p* < 0.05).

**Figure 7 plants-12-02232-f007:**
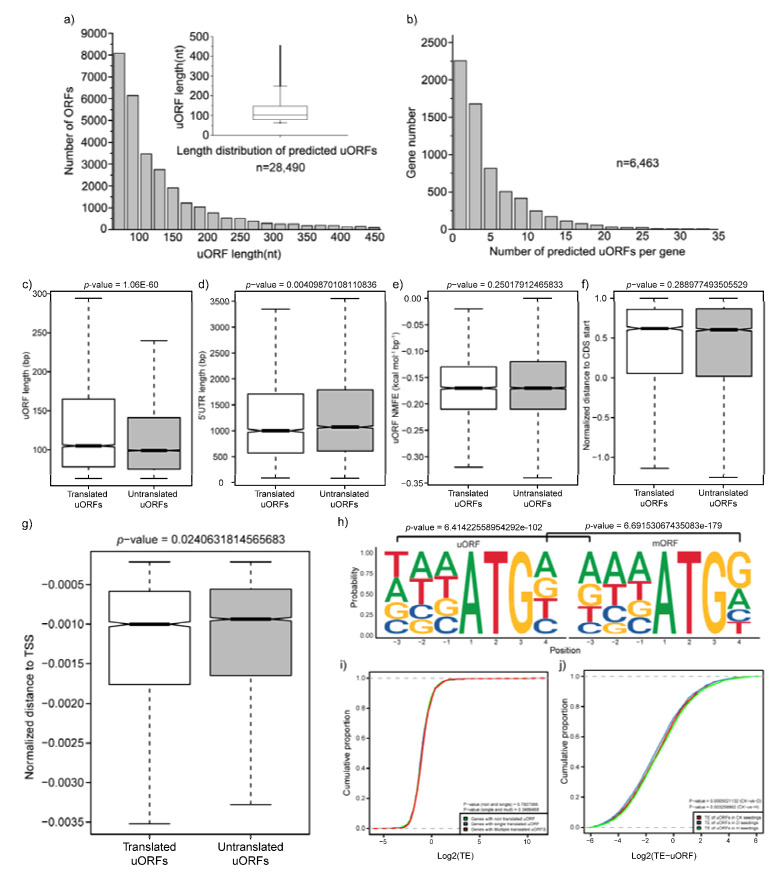
Prediction and characteristics of uORFs. (**a**) Length distribution of predicted uORFs; (**b**) number of predicted uORFs in each gene; (**c**) comparison of uORF length between translated and untranslated ones in the CK samples; (**d**) comparison of 5′UTR length between translated and untranslated ones in the CK samples; (**e**) comparison of uORFs NMFE between translated and untranslated ones in the CK samples; (**f**) comparison of normalized distance to CDS start between translated and untranslated ones in the CK samples; (**g**) comparison of normalized distance to the transcription start site (TSS) between translated and untranslated ones in the CK samples. *p*-values were tested by Student’s *t* test. (**h**) Kozak sequences of uORFs and main ORFs. The statistical significance was determined using Student’s *t* tests. (**i**) Comparison of translational efficiency among genes with no, one, and multiple translated uORFs in the CK samples; (**j**) comparison of TE of uORFs among CK, D, and H samples. *p*-values in (**i**,**j**) were calculated by Kolmogorov–Smirnov tests.

## Data Availability

The data in this manuscript are available from the corresponding author upon reasonable request.

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
