# Peer review of "Dynamic Translational Landscape Revealed by Genome-Wide Ribosome Profiling under Drought and Heat Stress in Potato"

_plants, 2023, doi:10.3390/plants12122232_

Round 1

Reviewer 1 Report

Dynamic translational landscape revealed by genome-wide ribosome profiling under drought and heat stress in potato

In the above manuscript authors  performed standard RNA-Seq and Ribo-Seq in potato seedlings under heat and drought stress and evaluated the early changes (3h) in transcription and translation and the association between these two key processes. Authors performed a thorough analysis of the data and provided all data-quality assessment data as well. A mass-spectrometry analysis to quantify the changes at the protein level would have added a greater strength to this study. However, the data  and the analysis will be of great value to the plant research community in general and potato researchers and breeders in particular. I would like the authors to address the following issues to make the manuscript acceptable for publication.

1.     The manuscript needs to be revised for English for readability and scientific accuracy. In many places authors arguments do not make any scientific sense especially in the introduction which was written very badly. Let me highlight some examples from the introduction. For instance, in the introduction authors are struggling to justify the reason why studying gene expression changes at the translational level is important. Also under introduction:

e.g. 1: Authors state Epigenetic changes have a direct influence in translation but does not provide any 

molecular explanation or evidence neither in the introduction nor in the discussion

e.g.2: There is no such thing called “monopolyribosomes”. Line 58-59 “Polyribosomes turn into 

monopolyribosomes when plants are subjected to external stress”. 

2.     It is well known that global translation is attenuated under various stresses. Authors did not explore this aspect in the data. Did the authors observed a general decrease in genes translated under stresses compared to control in general? Are there specific sets of transcripts differentially translated under stresses? Are they enriched for specific pathways or suggesting certain pathways? Present the categpries A-I (Figure 3) authors presented in the light of this analysis.

3.     CK, D and H stressed potato seedlings in line 88: Abbreviation used without introducing them and only introduced later in methods. Abbreviations need to be introduced before use.

4.     Labelling figure panels and matching the figure caption in confusing or there are errors. Particularly,

-       Figure 2b and c were switched in the caption

-       Figure 7b: X-axis says "predicted" and the figure caption says "identified". Which one is correct? 

-       Figure 7e is free energy? Follow the same pattern in the figure caption. e.g. "a)figure panel 'a' description or "figure panel 'a' description (a)". Don't mix.

5.     Under methods authors did not cite many of the original publication for the tools that they used. It is ethically only correct to cite. e.g. fastp, DESeq2, Bowtie2 etc.

6.     I was unable to find the dataset PRJNA925814. Please provide a link for reviewers to check whether the data is submitted satisfactorily.

Author Response

In the above manuscript authors performed standard RNA-Seq and Ribo-Seq in potato seedlings under heat and drought stress and evaluated the early changes (3h) in transcription and translation and the association between these two key processes. Authors performed a thorough analysis of the data and provided all data-quality assessment data as well. A mass-spectrometry analysis to quantify the changes at the protein level would have added a greater strength to this study. However, the data and the analysis will be of great value to the plant research community in general and potato researchers and breeders in particular. I would like the authors to address the following issues to make the manuscript acceptable for publication.

  1. The manuscript needs to be revised for English for readability and scientific accuracy. In many places authors arguments do not make any scientific sense especially in the introduction which was written very badly. Let me highlight some examples from the introduction. For instance, in the introduction authors are struggling to justify the reason why studying gene expression changes at the translational level is important. Also under introduction:

Response: Regarding the issue of language readability, we have asked experts to further revise and polish it.

e.g. 1: Authors state Epigenetic changes have a direct influence in translation but does not provide any molecular explanation or evidence neither in the introduction nor in the discussion.

Response: In this part, we want to introduce that except transcriptional regulation, plants also have post-transcriptional and other epigenetic regulation methods to cope with environmental stress when coping with environmental stress. Therefore, this study uses translatome and transcriptome sequencing technology to reveal that potatoes respond to high temperature and drought stress through translational regulation.

e.g.2: There is no such thing called “monopolyribosomes”. Line 58-59 “Polyribosomes turn into monopolyribosomes when plants are subjected to external stress”.

Response: We modified “monopolyribosomes” to “mono-polyribosomes” in our new MS.

  1. It is well known that global translation is attenuated under various stresses. Authors did not explore this aspect in the data. Did the authors observed a general decrease in genes translated under stresses compared to control in general? Are there specific sets of transcripts differentially translated under stresses? Are they enriched for specific pathways or suggesting certain pathways? Present the categpries A-I (Figure 3) authors presented in the light of this analysis.

Response: Yes, in total, 597 and 661 genes were significantly differentially expressed at translational level with no significant changes in transcription level for drought (Figure 3a) and heat (Figure 3c) stress, respectively. Nine pathways including biosynthesis of secondary metabolites, biosynthesis of amino acids and carbon metabolism and three pathways including ribosome, carbon metabolism, fructose and mannose metabolism were enriched in quadrant D after drought (Figure 3b) and heat (Figure 3d) stress, respectively. While no significantly pathway was enriched in group B and H in both drought and heat stress.

  1. CK, D and H stressed potato seedlings in line 88: Abbreviation used without introducing them and only introduced later in methods. Abbreviations need to be introduced before use.

Response: This has been modified in new MS.

  1. Labelling figure panels and matching the figure caption in confusing or there are errors. Particularly,

-       Figure 2b and c were switched in the caption

Response: This has been modified in new MS.

-       Figure 7b: X-axis says "predicted" and the figure caption says "identified". Which one is correct?

Response: This has been modified in new MS.

      Figure 7e is free energy? Follow the same pattern in the figure caption. e.g. "a) figure panel 'a' description or "figure panel 'a' description (a)". Don't mix.

Response: This has been modified in new MS.

  1. Under methods authors did not cite many of the original publication for the tools that they used. It is ethically only correct to cite. e.g. fastp, DESeq2, Bowtie2 etc.

Response: New cited papers were added in new MS as followed: 46) DESeq2: Three Differential Expression Analysis Methods for RNA Sequencing: limma, EdgeR, DESeq2; 47) fastp: fastp: an ultra-fast all-in-one FASTQ preprocessor; 48) Bowtie2: Langmead B, Salzberg S L. Fast gapped-read alignment with Bowtie 2[J]. Nature methods, 2012, 9(4): 357.

  1. I was unable to find the dataset PRJNA925814. Please provide a link for reviewers to check whether the data is submitted satisfactorily.

Response: Please find the raw data under the following link: https://dataview.ncbi.nlm.nih.gov/object/PRJNA925814?reviewer=77c1kksqdppqh74ql5h9a7sm9.

Reviewer 2 Report

I am very impressed by the presented manuscript, its relevance and high scientific value. In changing climate conditions, drought and heat stress are one of the main factors limiting crop growth and development. Therefore, it is of great importance for crop resistance breeding to investigate the regulatory network of gene expression regulatory network in response to abiotic stress. So far, the molecular mechanism of potato response to environmental changes at the translational level is not fully clear. In the present study, the authors elucidate the mechanism of potato regulatory response to high temperature and drought by combining translation omics and transcript omics. The obtained results provide new information about the potential transcriptional regulatory network in response to drought and heat stress, which in turn provides a new direction for improving tolerance to these stress factors.

In conclusion, I have no notes to the article thus presented, and propose that it be accepted as it is.

Author Response

Thanks!